# DUAL-MODULE INFERENCE FOR EFFICIENT RECURRENT NEURAL NETWORKS

## ABSTRACT

Using Recurrent Neural Networks (RNNs) in sequence modeling tasks is promising in delivering high-quality results but challenging to meet stringent latency requirements because of the memory-bound execution pattern of RNNs. We propose a *big-little* dual-module inference to dynamically skip unnecessary memory access and computation to speedup RNN inference. Leveraging the error-resilient feature of nonlinear activation functions used in RNNs, we propose to use a lightweight *little* module that approximates the original RNN layer, which is referred to as the *big* module, to compute activations of the insensitive region that are more error-resilient. The expensive memory access and computation of the *big* module can be reduced as the results are only used in the sensitive region. Our method can reduce the overall memory access by 40% on average and achieve 1.54x to 1.75x speedup on CPU-based server platform with negligible impact on model quality.

## 1 INTRODUCTION

Recurrent Neural Networks (RNNs) play a critical role in many natural language processing (NLP) tasks, such as machine translation (Bahdanau et al., 2014; Wu et al., 2016), speech recognition (Graves et al., 2013; He et al., 2019), and speech synthesis (Wang et al., 2017), owing to the capability of modeling sequential data. These RNN-based services deployed in both data-center and edge devices often process inputs in a streaming fashion, which demands a real-time interaction. For instance, in cloud-based translation tasks, multiple requests need to be served with very stringent latency limit, where inference runs concurrently and individually (Park et al., 2018). For on-device speech recognition as an automated assistant, latency is the primary concern to pursue a fast response (He et al., 2019).

However, serving RNN-based models in latency-sensitive scenarios is challenging due to the low data reuse, and thus low resource utilization as memory-bound General Matrix-Vector multiplication (GEMV) is the core compute pattern of RNNs. Accessing weight matrix from off-chip memory is the bottleneck of GEMV-based RNN execution as the weight data almost always cannot fit in on-chip memory. Moreover, accessing weights repeatedly at each time-step, especially in sequence-to-sequence models, makes the memory-bound problem severer. Subsequently, the on-chip computing resources would be under-utilized. Although batching is a walk-around for low-utilization, using a large batch size is not favored in latency-sensitive scenarios such as speech recognition and translation.

In essence, the RNN inference is not a simple GEMV. With non-linearity followed the GEMV operation as the activation functions, the RNN inference operation is "activated" GEMV. These nonlinear activation functions as used in neural networks bring error resilience. As shown in Figure 1, $sigmoid$ and $tanh$ functions in Gated RNNs such as Long Short-Term Memory (LSTM) (Hochreiter & Schmidhuber, 1997) and Gated Recurrent Unit (GRU) (Cho et al., 2014) have insensitive regions – green shaded regions – where the outputs are saturated and resilient to errors in pre-activation accumulated results. In other words, not all computations in RNNs need to be accurate. Can we leverage this error resilience in RNNs to reduce the memory access and eventually achieve speedup?

To this end, we propose a *big-little* dual-module inference that regarding the original RNN layer as the *big* module, and use a parameterized *little* module to approximate the *big* module to help reduce redundant weight accesses. The philosophy of dual-module inference is using approximated results computed by the memory-efficient *little* module in the insensitive region, and using accurate

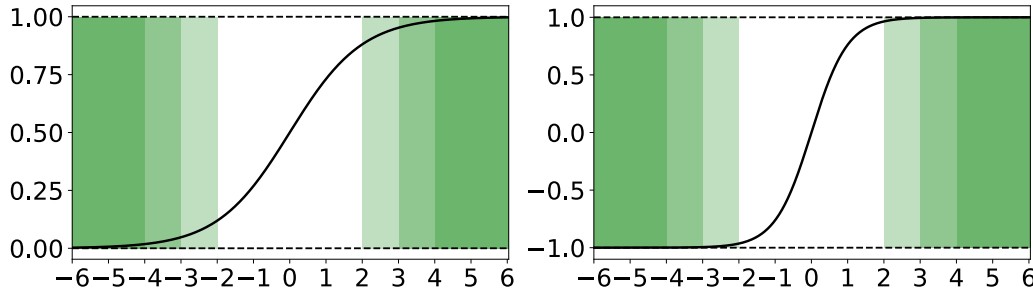

Figure 1: Insensitive (green shaded) and sensitive (white) regions of $sigmoid$ (left) and $tanh$ (right) nonlinear functions.

results computed by the memory-intensive *big* module in the sensitive region. For this reason, the final outputs are the mixture of the *big-little* module. With the memory-efficient *little* module computes for the insensitive region, we can reduce the expensive data access and computation of the *big* module and thus reduce overall memory access and computation cost. The (in)sensitive region is dynamically determined using the *little* module results. Because of the error resilience, using approximated results in the insensitive region has a negligible impact on the overall model quality but creates a significant acceleration potential.

Given the trade-off between accuracy and efficiency, the *little* module needs to be sufficiently accurate while being as much lightweight as possible. To achieve this, we first use a dimension reduction method – random projection – to reduce the parameter size of the *little* module and thus reducing data accesses. Then, we quantize the weights of the *little* module to lower the overhead further. Because we only need the *little* module outputs in the insensitive region that is error-resilient, we can afford aggressively low bit-width. Compared with common sparsification schemes, our hybrid approach avoids indexing overheads and therefore successfully achieves practical speedup.

We evaluate our method on language modeling and neural machine translation using RNN-based models and measure the performance, i.e., wall-clock execution time, on CPU-based server platform. With overall memory access data reduced by 40% on average, our method can achieve 1.54x to 1.75x speedup with negligible impact on model quality.

## 2 MOTIVATION

In this section, we discuss the error resilience of RNNs. As shown in Fig. 1, the nonlinear activation functions – $sigmoid$ and $tanh$ – have insensitive regions where the output activations are resilient to errors introduced in pre-activation accumulation results. We take a single LSTM layer for language modeling over PTB dataset as an illustrative example. The baseline perplexity (PPL) is 80.64. We consider two cases: adding a random error vector under norm distribution into the pre-activation accumulation results in the sensitive regions of four gates; adding errors to the insensitive regions. We separate the (in)sensitive regions by 50% based on the activation magnitude.

As listed in Table 1, we report the PPL on the testing set and the average cosine similarity between the activations of the baseline model and the error-introduced model. Before applying the nonlinear activation functions, the cosine similarity of two cases – adding errors in the sensitive region or the insensitive region – are in the same level. However, we observe that after the nonlinear gates, the cosine similarity in the insensitive case is much closer to one (i.e., fewer output errors) than that in

Table 1: Comparison of adding random errors to the sensitive or insensitive region of LSTM gates.

| Case | Cosine similarity before gate | | | | Cosine similarity after gate | | | | PPL |
|---|---|---|---|---|---|---|---|---|---|
| | input | forget | cell | output | input | forget | cell | output | |
| Sensitive | 0.953 | 0.859 | 0.952 | 0.932 | 0.934 | 0.946 | 0.882 | 0.940 | 85.70 |
| Insensitive | 0.944 | 0.929 | 0.943 | 0.947 | 0.968 | 0.987 | 0.969 | 0.977 | 81.79 |

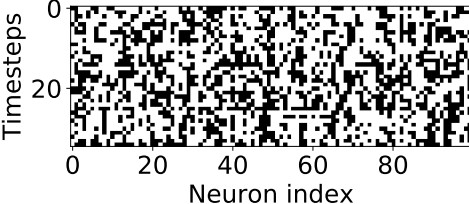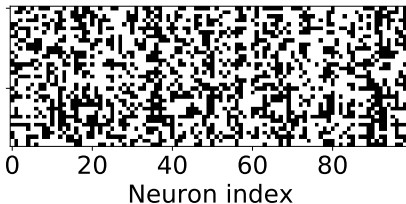

Figure 2: Dynamic region distribution across timesteps and inputs. The white and black colors denote neurons in the insensitive and sensitive regions, respectively. The left and right patterns are from different inputs.

the sensitive case. We further compare the PPL of these two cases, and we observe that introducing errors in the insensitive region causes little quality degradation.

The selection of which neurons should be in the (in)sensitive region is dynamic and input-dependent, which can be seen in Figure 2. Unlike the static weight sparsity that we can prune the unused connections offline in advance, the dynamic region speculation requires a very lightweight criterion for real-time processing. Taking all these into account, we propose a dual-model inference method that efficiently determines (in)sensitive region and significantly saves the memory access and computational cost.

## 3 APPROACH

Firstly, we explain the dual-module inference by taking a fully-connected (FC) layer as an example and then extend it to LSTM and GRU. For an FC layer with unit batch size, the operation is typically formulated as $\boldsymbol{a} = \varphi(\boldsymbol{y}), \boldsymbol{y} = \boldsymbol{W}\boldsymbol{x} + \boldsymbol{b}$, where $\boldsymbol{W}$ is a weight matrix ($\boldsymbol{W} \in \mathbb{R}^{n \times d}$), $\boldsymbol{x}$ is an input vector ($\boldsymbol{x} \in \mathbb{R}^d$), $\boldsymbol{b}$ is a bias vector ($\boldsymbol{b} \in \mathbb{R}^n$), $\boldsymbol{a}$ is an activated output vector ($\boldsymbol{a} \in \mathbb{R}^n$), and $\varphi$ is an activation function. The core computation is matrix-vector multiplication (GEMV), i.e., $\boldsymbol{W}\boldsymbol{x}$. Both the amount of computation and memory access are $O(nd)$; therefore, it is memory-bounded since the operation intensity is $O(1)$ according to the Roofline model analysis (Williams et al., 2009). Accessing weights from the off-chip memory is the bottleneck in terms of both the latency and energy.

### 3.1 OVERVIEW OF DUAL-MODULE PHILOSOPHY

Our work aims at reducing the memory access of weight matrices for GEMV-based RNN inference. We show in Section 2 that not all values in $\boldsymbol{y}$ need accurate computation, and those that belong to the insensitive region can afford some level of approximation. In other words, we only need accurate computation and expensive memory access in the sensitive region of $\boldsymbol{y}$ and skip computation and memory access to weights that contribute to the insensitive region of $\boldsymbol{y}$. With that, we still need approximated results in the insensitive region. Therefore, we propose to learn a lightweight *little* module from the original trained layer, here we refer the original layer as the *big* module. Essentially, our *little* module is executed in a low-dimensional and low-precision space, thus termed as $LL$ module; by contrast, the original *big* module with high dimension and high precision is called $HH$ module. Let the outputs from these two modules be $\boldsymbol{y}^{LL}$ and $\boldsymbol{y}^{HH}$, respectively. If the $LL$ module approximates the $HH$ module well, the final output vector – a mixture of results from the $HH$ and the $LL$ modules – can be assembled by

$$\boldsymbol{y} = \boldsymbol{y}^{HH} \odot \boldsymbol{m} + \boldsymbol{y}^{LL} \odot (1 - \boldsymbol{m}) \tag{1}$$

where $\boldsymbol{m} \in \{0, 1\}^n$ is a binary mask vector for the output switching. $m_i$ equals 1 in the sensitive region while it switches to 0 in the insensitive region. The overall saving comes from skipping memory access to the *big* module while paying the overhead of accessing and computing of the *little* module.

## 3.2 CONSTRUCT THE $LL$ MODULE

As the $HH$ module is the original pre-trained layer, we only need to construct the $LL$ module. Delivering a lightweight *little* module at inference time is crucial to achieving real wall-clock time speedup. As discussed earlier, the sparsification method usually suffers from severe indexing overheads; therefore, we turn to other approaches. In this work, we propose a hybrid compression with dimension reduction and data quantization to keep the *little* module as efficient as possible in computation and storage. The low dimension and low precision give birth to the desired $LL$ module. We emphasize two objects that should be reached in the design of $LL$ module: (1) much lower computation and memory overheads than the $HH$ module; (2) approximating the outputs of $HH$ module accurately.

First, we introduce sparse random projection to reduce the dimension of $x$ from $\mathbb{R}^d$ to $\mathbb{R}^k$ where $k \ll d$. Subsequently, the parameter size of the $LL$ module is $O(nk)$, which is much smaller compared with the parameter size $O(nd)$ of the $HH$ module. Random projection is a common technique for dimension reduction that preserves distances in Euclidean space (Achlioptas, 2003; Bingham & Mannila, 2001; Li et al., 2006; Liu et al., 2019).

The dimension reduction step can be formulated as

$$\boldsymbol{x}^{LL} = \boldsymbol{P}\boldsymbol{x}^{HH} \tag{2}$$

where $\boldsymbol{P}$ is a sparse random matrix ($\boldsymbol{P} \in \frac{1}{\sqrt{3}} \cdot \{-1, 0, 1\}^{k \times d}$, the probability of $P_{ij}$ being $-1$, $0$, and $1$ is $\frac{1}{6}$, $\frac{2}{3}$, and $\frac{1}{6}$, respectively). Note that $k$ is configurable according to actual needs to balance the accuracy loss and inference cost. We choose the value of $k$ according to Achlioptas (2003):

$$k = \frac{4 log n}{\epsilon^2/2 - \epsilon^3/3} \tag{3}$$

where $n$ is the number of rows in $\boldsymbol{W}$ and $\epsilon$ is a real number in $(0, 1)$.

Second, after the dimension reduction, we quickly construct a lightweight *little* module in the low-dimensional space to approximate the pre-trained *big* module. The parameters of the latter (i.e., $\boldsymbol{W}^{HH}$ and $\boldsymbol{b}^{HH}$) are kept frozen while the parameters of the former (i.e., $\boldsymbol{W}^{LL}$ and $\boldsymbol{b}^{LL}$) are updated by stochastic gradient descent (SGD) to minimize the following loss function:

$$L = \frac{1}{S} \sum_s ||\boldsymbol{y}^{HH} - \boldsymbol{y}^{LL}||_2^2 = \frac{1}{S} \sum_s ||(\boldsymbol{W}^{HH}\boldsymbol{x}^{HH} + \boldsymbol{b}^{HH}) - (\boldsymbol{W}^{LL}\boldsymbol{x}^{LL} + \boldsymbol{b}^{LL})||_2^2 \tag{4}$$

where $S$ is the mini-batch size. Essentially, for each pair of *big-little* modules, we apply linear regression on the *little* module to approximate the function of the *big* module and optimize the mean square error of the two. Apparently, the parameter size of $\boldsymbol{W}^{LL}$ is $O(nk)$, much smaller than the original weight $\boldsymbol{W}^{HH}$ of $O(nd)$ in the high-dimensional space. Even if further considering the projection cost of $O(kd)$, the overhead is still much lower than the vanilla inference. In this way, the memory-bound issue in GEMV-based models can be greatly alleviated; the computational complexity is also reduced. The SGD overhead for constructing the above module can be amortized by the pattern of "construct-once-inference-forever".

Finally, based on the constructed low-dimensional module, we also apply data quantization technique to reduce the parameter precision. Data quantization can further shrink the storage space of $LL$ parameters due to the shorter bit-width. The input $\boldsymbol{x}$ is also quantized during run-time to reduce the computation cost. In our design, we apply one-time uniform quantization on $\boldsymbol{W}^{LL}$ to avoid complicated calculations. Although some other accurate quantization methods are available as well, we find that one-time quantization works well in our dual-module inference given in Equation (1). This error tolerance is benefit from the fact that the computation in the insensitive region has a small influence on the final outputs.

## 3.3 DETERMINE THE INSENSITIVE REGION

The dual-module inference relies on a binary mask $\boldsymbol{m}$ to switch between outputs of the "accurate & costly" $HH$ module and the "approximated & efficient" $LL$ module. Hence, the generation of $\boldsymbol{m}$ is a crucial factor to control the overall performance by adjusting the trade-off between accuracy and efficiency. Thanks to the saturation region of the nonlinear activation functions in RNNs, such

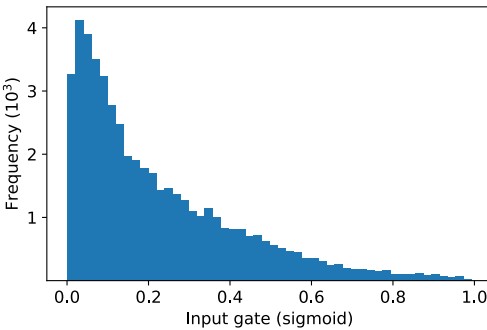 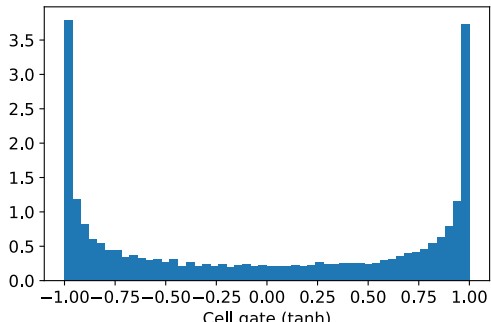

Figure 3: Output activation distribution of $sigmoid$ and $tanh$ gates as in LSTM.

as $sigmoid$ and $tanh$, we observe a unipolar or bipolar distribution of their outputs, as depicted in Figure 3. This affords two excellent opportunities: (1) It is possible to remove the majority of the computation and access from the costly $HH$ module by setting the peak areas in Figure 3 as insensitive regions; (2) The saturation output values in those regions such as near 0 in $sigmoid$ and near $\pm 1$ in $tanh$ additionally allow inaccurate computations because the outputs are insensitive to approximated values. According to the above observations and analysis, we design a specific criterion for each activation function. In particular, they are governed by

$$
\begin{cases}
sigmoid: \ if \ y_i^{LL} > \theta_{sigmoid}, m_i = 1; \ otherwise, \ m_i = 0 \\
tanh: \ if \ \theta_{tanh}^- < y_i^{LL} < \theta_{tanh}^+, m_i = 1; \ otherwise, \ m_i = 0
\end{cases}
\tag{5}
$$

where $\theta_{sigmoid} > 0$, $\theta_{tanh}^- < 0$, and $\theta_{tanh}^+ > 0$ are constant thresholds. Note that these thresholds can be searched to a target insensitive ratio using validation dataset or be tuned at run-time that acts as a knob for accuracy-efficiency trade-off.

### 3.4 Overview of Dual-Module Inference Algorithm

The overall implementation is provided in Algorithm 1. After the construction of the $LL$ model, the consequent dual-module inference needs five steps: (1) Dimension reduction and data quantization for each dynamical input $\boldsymbol{x}$ as $\boldsymbol{x}_Q^{LL} = Q(\boldsymbol{P}\boldsymbol{x}^{HH})$ where $Q(\cdot)$ is a quantization function; (2) Obtain the approximated output $\boldsymbol{y}^{LL}$ by performing $\boldsymbol{y}^{LL} = \varphi(\boldsymbol{W}_Q^{LL}\boldsymbol{x}_Q^{LL} + \boldsymbol{b}_Q^{LL})$ where $\boldsymbol{W}_Q^{LL}$ & $\boldsymbol{b}_Q^{LL}$ are stored quantized parameters; (3) Calculate the switching mask vector $\boldsymbol{m}$ according to Equation (5); (4) Obtain a faction of actual output $\boldsymbol{y}^{HH}$ by performing $y_i^{HH} = \varphi(\boldsymbol{W}[i,:]^{HH}\boldsymbol{x}^{HH} + b_i^{HH})$ if $m_i = 1$; (5) Produce the final output $\boldsymbol{y}$ according to the assembling in Equation (1).

---

**Algorithm 1:** Dual-module Inference Algorithm

**Data:** $HH$ module parameters: $\boldsymbol{W}^{HH}$, $\boldsymbol{b}^{HH}$; quantized $LL$ module parameters: $\boldsymbol{W}_Q^{LL}$ and $\boldsymbol{b}_Q^{LL}$;
        thresholds $\theta$s to determine $\boldsymbol{m}$; random projection matrix $\boldsymbol{P}$; current input $\boldsymbol{x}^{HH}$

**Result:** Final output $\boldsymbol{y}$

1 Step 1: $\boldsymbol{x}_Q^{LL} = Q(\boldsymbol{P}\boldsymbol{x}^{HH})$;

2 Step 2: $\boldsymbol{y}^{LL} = \varphi(\boldsymbol{W}_Q^{LL}\boldsymbol{x}_Q^{LL} + \boldsymbol{b}_Q^{LL})$;

3 Step 3: Generating $\boldsymbol{m}$ according to Equation (5);

4 Step 4-5: **foreach** $m_i \in \boldsymbol{m}$ **do**

5      **if** $m_i == 1$ **then** $y_i = y_i^{HH} = \varphi(\boldsymbol{W}[i,:]^{HH}\boldsymbol{x}^{HH} + b_i^{HH})$;

6      **else** $y_i = y_i^{LL}$;

7 **end**

---

### 3.5 Apply to Recurrent Neural Networks

We discuss how to apply the proposed dual-module inference for an FC layer to RNNs, including LSTM and GRU. We will explain the LSTM implementation for illustration, while the extension to

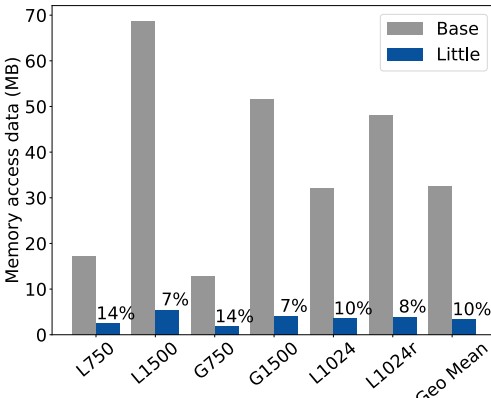 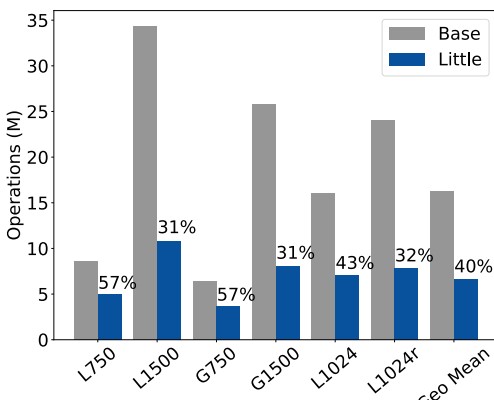

Figure 4: Comparison of memory access data and operations between baseline layers and the *little* module of dual-module enhanced layers. *L750* indicates single-layer LSTM with 750 hidden units; *G* is short for GRU; *L1024r* indicates LSTM with residual input.

GRU is quite straightforward. The dynamics of an LSTM cell can be described as

$$
\begin{cases}
\boldsymbol{f}(t) = \sigma(\boldsymbol{b}_f + \boldsymbol{W}_{fx}\boldsymbol{x}(t) + \boldsymbol{W}_{fh}\boldsymbol{h}(t-1)) \\
\boldsymbol{i}(t) = \sigma(\boldsymbol{b}_i + \boldsymbol{W}_{ix}\boldsymbol{x}(t) + \boldsymbol{W}_{ih}\boldsymbol{h}(t-1)) \\
\boldsymbol{o}(t) = \sigma(\boldsymbol{b}_o + \boldsymbol{W}_{ox}\boldsymbol{x}(t) + \boldsymbol{W}_{oh}\boldsymbol{h}(t-1)) \\
\boldsymbol{g}(t) = \theta(\boldsymbol{b}_g + \boldsymbol{W}_{gx}\boldsymbol{x}(t) + \boldsymbol{W}_{gh}\boldsymbol{h}(t-1)) \\
\boldsymbol{c}(t) = \boldsymbol{c}(t-1) \odot \boldsymbol{f}(t) + \boldsymbol{g}(t) \odot \boldsymbol{i}(t) \\
\boldsymbol{h}(t) = \theta(\boldsymbol{c}(t)) \odot \boldsymbol{o}(t)
\end{cases}
\tag{6}
$$

where $\boldsymbol{f}, \boldsymbol{i}, \boldsymbol{o}$ are the states of forget, input, and output gate, respectively, and $\boldsymbol{g}$ is the input activation. Each of them has its own bias vector and weight matrices. $\boldsymbol{c}$ and $\boldsymbol{h}$ are the cellular and hidden states of the hidden layer, respectively. $\sigma(\cdot)$ and $\theta(\cdot)$ are $sigmoid$ function and $tanh$ function, respectively.

The computation of each gate is similar to an FC-like layer; therefore, Algorithm 1 still holds. The first difference is the two GEMV computations in each gate; we apply dimension reduction, construction of the $LL$ module, and data quantization on both GEMV computations. The second difference is that there is an additional temporal dimension in RNNs. We should guarantee the approximation performance of the $LL$ module at all time steps. Taking the forget gate as an example, the linear map works for both $\boldsymbol{x}^{LL}(t) = \boldsymbol{P}_x\boldsymbol{x}^{HH}(t)$ and $\boldsymbol{h}^{LL}(t-1) = \boldsymbol{P}_h\boldsymbol{h}^{HH}(t-1)$. The loss function for constructing the $LL$ module is slightly modified to

$$
L = \frac{1}{ST}\sum_s\sum_t ||(\boldsymbol{b}_f^{HH} + \boldsymbol{W}_{fx}^{HH}\boldsymbol{x}^{HH}(t) + \boldsymbol{W}_{fh}^{HH}\boldsymbol{h}^{HH}(t-1)) - (\boldsymbol{b}_f^{LL} + \boldsymbol{W}_{fx}^{LL}\boldsymbol{x}^{LL}(t) + \boldsymbol{W}_{fh}^{LL}\boldsymbol{h}^{LL}(t-1))||_2^2.
\tag{7}
$$

Here the minimization considers not only $S$ training samples in each mini-batch but also $T$ time steps. The data quantization, switching mask (i.e., $\boldsymbol{m}$) generation, and output assembling is the same as Algorithm 1 describes. Applying to other gates is similar so we do not discuss them to avoid repetition. Note that the input $\boldsymbol{x}$ and hidden state $\boldsymbol{h}$ can have different sizes, termed as $d_x$ and $d_h$, respectively. For simplicity, we set $P_x \in \mathbb{R}^{k \times d_x}$ and $P_h \in \mathbb{R}^{k \times d_h}$ to let $\boldsymbol{x}^{LL}$ and $\boldsymbol{h}^{LL}$ to the same length $k$. For the $\boldsymbol{g}$ gate with $tanh$ function, we set $|\theta_{tanh}^-| = |\theta_{tanh}^+|$ also for simplicity; however, different magnitudes are allowed.

## 3.6 SAVING AND OVERHEAD ANALYSIS

The target of our dual-module inference method is to reduce the expensive off-chip memory access of the *big* module with the help of the *little* module. We introduce an insensitive ratio as the number of outputs using the *little* module results over entire outputs. The ratio can be interpreted as the zero ratio in mask $\boldsymbol{m}$ as in Equation 1. In other words, the higher insensitive ratio will have less memory access to the *big* module. For example, obtaining a ratio of 50% results in reducing 50% of weight matrix accessing in a GEMV operation. The choice of accurate ratio determines the model inference quality, and it is a knob to trade-off model inference quality vs. latency at run-time.

The overhead of dual-module inference is small due to the use of dimension reduction and quantization. When choosing reduced dimension $k$ and low-precision bit-width of the *little* module, we use Equation 3 with $\epsilon = 0.5$ and INT8 quantization by default. We also explore different levels of dimension reduction and quantization in Section 4.3 and Section 4.4. As shown in Figure 4, we compare memory access data and operations between the single-module – the base case – and the *little* module of dual-module inference using a set of LSTM and GRU layers. On average, the *little* module accounts 10% storage overhead and 40% operation overhead compared with the base case. Note that we count the number of operations in Figure 4 regardless of precision; and the *little* module computation overhead can be further reduced using low-precision compute kernel as we used in performance evaluation.

## 4 EVALUATION

We first evaluate the model inference quality and execution time under different insensitive ratio and then conduct two sensitivity studies on dimension reduction and quantization.

Our method is evaluated on CPU-based server platform (Intel(R) Xeon(R) CPU E5-2698 v4) as most inference workloads run on CPUs (Park et al., 2018). We use PyTorch to train the *little* module and evaluate inference quality. The baseline implementation is the PyTorch CPU version with Intel MKL (version 2019.4) as the back-end BLAS kernel library. Our custom kernel implementation uses a multi-threaded MKL dot-product kernel at BLAS level-1 to compute the *big* module instead of BLAS level-2 or level-3 kernels. The kernel-wise performance is measured as wall-clock time and averaged with 1000 runs, assuming cold cache at the execution of each RNN cell representing the real-world cases, for example in the decoder of seq2seq model.

We first evaluate our method on single-layer LSTM & GRU used in language modeling tasks and then on multi-layer stacked LSTM in GNMT model used in machine translation tasks – a standard benchmark model for inference as in MLPerf [1]. We train the *little* module while freezing the parameters of the *big* module, and we use the same training set and validation set to run SGD optimization.

### 4.1 LANGUAGE MODELING

We first evaluate our method on single-layer LSTMs/GPUs. Our implementations are adapted from the word-level language modeling example from PyTorch with same hyper-parameters to train baseline models. We report word-level perplexity (PPL) as the measure of model quality. As listed in Table 2, the baseline LSTM model achieves 80.64 PPL at the latency of 1.477ms. Then, we varying the insensitive ratio to show the quality-performance trade-off; the larger insensitive ratio indicates more results are from the *little* module and less memory access to compute the *big* module. As we increase the insensitive ratio, we observe the degradation of quality as the perplexity increases during a gradual reduction in execution time. When the insensitive ratio is 50%, the perplexity is slightly increased to 81.36, which is negligible in language modeling tasks, while the inference speedup is 1.67x.

We observe a similar quality-performance trade-off for LSTM with 750 hidden units. Comparing the case of base LSTM with 750 hidden units with dual-module LSTM with 1500 hidden units and 50% insensitive ratio, although the memory access reduction is at the same level, our proposed dual-module approach achieves much better model quality because we kept the expressive power of a larger LSTM layer.

We further report the results using single-layer GRU on word-level language modeling tasks as in Table 3. Using dual-module inference on GRUs expresses the similar quality-performance trade-off as of LSTMs. Our dual-module method is generally applicable to both LSTMs and GRUs.

### 4.2 NEURAL MACHINE TRANSLATION

Given the promising results on language modeling, we further investigate Neural Machine Translation (NMT), which is a promising end-to-end learning approach for automated translation (Wu

---

[1]https://mlperf.org/inference-overview/

Table 2: LSTM perplexity and execution time (ms).

| Insensitive | hidden size: 1500 | | | | hidden size: 750 | | | |
|---|---|---|---|---|---|---|---|---|
| ratio | PPL | Diff. | Time | Speedup | PPL | Diff. | Time | Speedup |
| Base | 80.64 | n/a | 1.477 | 1.00x | 84.32 | n/a | 0.546 | 1.00x |
| 10% | 80.72 | -0.08 | 1.315 | 1.12x | 84.42 | -0.10 | 0.448 | 1.22x |
| 30% | 80.56 | 0.08 | 1.095 | 1.35x | 84.43 | -0.11 | 0.415 | 1.32x |
| 50% | 81.36 | -0.72 | 0.885 | 1.67x | 84.29 | 0.03 | 0.342 | 1.60x |
| 70% | 87.48 | -6.83 | 0.641 | 2.30x | 84.89 | -0.57 | 0.287 | 1.90x |
| 90% | 109.37 | -28.73 | 0.380 | 3.89x | 88.44 | -4.12 | 0.216 | 2.53x |

Table 3: GRU perplexity and execution time (ms).

| Insensitive | hidden size: 1500 | | | | hidden size: 750 | | | |
|---|---|---|---|---|---|---|---|---|
| ratio | PPL | Diff. | Time | Speedup | PPL | Diff. | Time | Speedup |
| Base | 85.48 | n/a | 1.182 | 1.00x | 89.64 | n/a | 0.466 | 1.00x |
| 10% | 85.62 | -0.14 | 1.024 | 1.15x | 89.81 | -0.17 | 0.383 | 1.22x |
| 30% | 86.01 | -0.53 | 0.869 | 1.36x | 89.63 | 0.01 | 0.334 | 1.40x |
| 50% | 88.73 | -3.25 | 0.726 | 1.63x | 89.69 | -0.05 | 0.302 | 1.54x |
| 70% | 98.09 | -12.61 | 0.545 | 2.17x | 92.51 | -2.87 | 0.284 | 1.64x |
| 90% | 122.75 | -37.27 | 0.350 | 3.38x | 102.37 | -12.73 | 0.198 | 2.35x |

et al., 2016). The base model [2] consists of a four-layer stacked LSTM in both the encoder and the decoder of the sequence-to-sequence modeling. We focus on the speedup of the decoder since it is the most memory intensive and the most time-consuming part ( 95%). The decoder has a four-layer unidirectional LSTM with hidden size 1024 with residual connections starting from the third layer, i.e., the input size of the third and fourth layer is 2048. Our experiments show de-tokenized BLEU score to measure the model inference quality on the public WMT16 English-German dataset. The baseline model obtains a BLEU score of 24.32.

We replace the LSTM layers in the decoder with our proposed dual-module-based LSTM layers. Similar to single-layer LSTM results, using the *little* module computed results in the insensitive region can reduce overall memory access while maintaining model quality. As listed in Table 4, our method can achieve imperceptible BLEU score degradation while speedup inference by 1.75x for the first two LSTM layers and 1.70x for the last two LSTM layers. When compromising more translation quality, i.e., decreasing the BLEU score by 2.4, our method can achieve more than 2x speedup.

### 4.3 DISCUSSION ON DIMENSION REDUCTION

Dimension reduction is an integral part of our dual-module inference method to reduce the number of parameters and memory footprint. Here, we study the impact of different levels of dimension reduction on the model quality and performance. We conduct experiments on language modeling using single-layer LSTM of 1500 hidden units. We quantize the *little* module to INT8 and reduce the hidden dimension from 1500 to three different levels, which are calculated by Sparse Random Projection. We fix the insensitive ratio to be 50% across this set of experiments. As we can see in Table 5, the higher dimension of the *little* module, the better approximation the *little* module can perform. For instance, when we reduce hidden size to 966 and quantize to INT8, the dual-module inference can achieve slightly better quality – PPL of 80.40 – and 1.37x speedup. More aggressive dimension reduction can further have more speedup at the cost of more quality degradation: hidden dimension reduced to 417 and 266 can have 1.67x and 1.71x speedup but increase PPL by 0.72 and 2.87, respectively.

We further show the overhead of performing the computation of the *little* module. As listed in the last three columns in Table 5, we measure the execution time of performing dimension reduction on

---

[2]From https://github.com/NVIDIA/DeepLearningExamples

Table 4: GNMT BLEU score and execution time (ms). (1024, 2048) indicates the hidden size is 1024 and the input size is 2048; similarly for (1024, 1024).

| Insensitive ratio | Quality | | (1024, 1024) | | (1024, 2048) | |
|---|---|---|---|---|---|---|
| | BLEU | Diff. | Time | Speedup | Time | Speedup |
| Base | 24.32 | n/a | 0.838 | 1.00x | 1.092 | 1.00x |
| 10% | 24.33 | 0.01 | 0.679 | 1.23x | 0.962 | 1.14x |
| 30% | 24.18 | -0.14 | 0.541 | 1.55x | 0.803 | 1.36x |
| 50% | 23.73 | -0.59 | 0.480 | 1.75x | 0.642 | 1.70x |
| 70% | 21.92 | -2.40 | 0.360 | 2.33x | 0.479 | 2.28x |
| 90% | 11.77 | -12.55 | 0.243 | 3.45x | 0.307 | 3.56x |

Table 5: Sensitivity study of dimension reduction.

| Dimension | PPL | Diff. | Time | Speedup | SRP | Little | Big |
|---|---|---|---|---|---|---|---|
| 1500 (baseline) | 80.64 | n/a | 1.477 | 1.00x | 0% | 0% | 100% |
| 966 ($\epsilon = 0.3$) | 80.40 | 0.24 | 1.076 | 1.37x | 8% | 14% | 44% |
| 417 ($\epsilon = 0.5$) | 81.36 | -0.72 | 0.885 | 1.67x | 4% | 8% | 47% |
| 266 ($\epsilon = 0.7$) | 83.51 | -2.87 | 0.866 | 1.71x | 3% | 5% | 46% |

inputs by Sparse Random Projection, computation of the *little* module, and computation of the *big* module; the execution time is normalized to the baseline case, i.e., the execution time of standard LSTM, to highlight the percentage of overheads. When the hidden dimension is reduced to 966, the overhead of the *little* module accounts 22% while the execution time of the *big* module is cut off by half [3]. In our experiments, we choose $\epsilon = 0.5$ as the default parameter in sparse random projection as it demonstrated good quality and speedup trade-off by our study. When further reducing the hidden dimension to 266, there is only a slight improvement on speedup compared with the hidden size of 417 in the *little* module, where the overhead of the *little* module is already small enough, but the quality dropped significantly.

## 4.4 DISCUSSION ON QUANTIZATION

Quantizing the weights of the *little* module is another integral part of keeping memory footprint small. We show different quantization levels the impact on model quality and parameter size. After training the *little* module, we can quantize its weights to lower precision to reduce the memory accessing on top of dimension reduction. As we can see in Table 6, more aggressive quantization leads to smaller parameter size that can reduce the overhead of computing the *little* module; on the other hand, the approximation of the *little* module is compromised by quantization. We can quantize the *little* module up to INT4 without significant quality degradation. Using lower precision would degrade the quality while decreasing the parameter size. For performance evaluation, we choose INT8 as the quantization level since we leverage off-the-shelf INT8 GEMM kernel in MKL. We expect more speedup once the *little* module overhead can be further reduced by leveraging INT4 compute kernels.

Table 6: Inference quality and parameter size comparison under different levels of quantization on the *little* module

| Precision | Base | FP32 | INT16 | INT8 | INT4 | INT2 | INT1 |
|---|---|---|---|---|---|---|---|
| Perplexity | 80.64 | 81.28 | 81.18 | 81.25 | 81.28 | 82.14 | 94.75 |
| Diff. | n/a | -0.64 | -0.54 | -0.61 | -0.64 | -1.50 | -14.11 |
| MSE | n/a | 0.408 | 0.425 | 0.444 | 0.465 | 0.573 | 3.337 |
| Param. size (MB) | 68.7 | 19.1 | 9.6 | 4.8 | 2.4 | 1.2 | 0.6 |

---

[3]We measured the execution time with multi-threading.

## 5 RELATED WORK

As we aim at the memory-bound problem of RNN-based inference applications, we limit the discussion on related work to RNN inference acceleration. Although we only evaluate our dual-module inference method on standard LSTMs/GRUs, we believe our method can be applied to many newly released sequence modeling networks (Shen et al., 2019; Wu et al., 2019) as we leverage the commonly observed error-resilience of non-linear activation functions.

### 5.1 MODEL COMPRESSION

Compressing DNN models via data quantization, weight sparsity, and knowledge distillation is promising to deliver efficient deployment for inference. Xu et al. (2018) propose a quantization method for RNNs where both weights and activations are quantized to binary or ternary. Wang et al. (2018) propose a hybrid ternary quantization method based on the different distributions of weights and activations.

Weight pruning, i.e., inducing weight sparsity, has been proposed to reduce the parameter size of a pre-trained model (Han et al., 2015b;a). While fine-grained pruning at element-wise could reduce the number of parameters (Narang et al., 2017; Zhu & Gupta, 2017; Dai et al., 2018), indexing non-zero weights causes extra memory cost and would offset the benefits of reducing parameter size; it is hard to gain practical acceleration on general-purpose hardware or need hardware specialization (Mao et al., 2017). Although structural pruning (Wen et al., 2017) and knowledge distillation (Polino et al., 2018) could achieve speedup, the applicability on more complicated tasks such NMT using large-scale dataset is unstudied; besides, those methods require extensive retraining via regularization that would increase the training cost and hard to find a solution.

Model compression would inevitably compromise the compressive power of RNNs. Our method, by no means, is supposed to replace model compression but provides an orthogonal approach to accelerate RNN inference. Using the analogy of knowledge distillation, we do not simply deploy a student network learned from the teacher network. Instead, we let the teacher network, applied with model compression or not, help with the student – the *little* module learned from the base module – and collaboratively perform inference with reduced memory access and computation.

### 5.2 COMPUTATION SKIPPING

Instead of model compression, many work propose to skip computations dynamically based on certain criterion. Bolukbasi et al. (2017) propose dynamic execution with layer-wise early exit. Zhang et al. (2018) leverage a special feature of LSTM that using threshold-based pruning on output gates and generate a mask, and then using the mask to skip computation as well as data access of masked-out neurons of the other three gates. Neil et al. (2017) utilize temporal input sparsity but need to enforce input similarity with threshold clipping. Campos et al. (2018) selectively skip updating the hidden states for some inputs. However, these work either depend on special cell structure or rely on the temporal similarity of inputs which is not evaluated on NLP tasks such as NMT. We are the first that propose a general and principled method to reduce memory access and computation of Gated RNNs, including both LSTMs and GRUs.

## 6 CONCLUSION

In this paper, we describe a *big-little* dual-module inference method to mitigate the memory-bound problem in serving RNN-based models under latency-sensitive scenarios. We leverage the error resilience of nonlinear activation functions by using the lightweight *little* module to compute for the insensitive region and using the *big* module with skipped memory access and computation to compute for the sensitive region. With overall memory access reduced by near half, our method can achieve 1.54x to 1.75x wall-clock time speedup without significant degradation on model quality.

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

## APPENDIX A  COMPARISON WITH WEIGHT PRUNING METHOD

We compare our proposed dual-module inference approach with the automated gradual pruning method (Zhu & Gupta, 2017), which is a popular pruning method with open implementation[4]. Firstly, compared with weight pruning, our method achieves better quality with practical speedup – 1.54x to 1.75x reduction on wall-clock time – on commodity CPUs while element-wise weight pruning requires specialized hardware to gain real speedup of computation given irregular sparsity. Moreover, our dual-module inference method can be further applied on top of pruned models to reduce execution time by reducing memory access.

Table 7: Comparison of our proposed dual-module inference (using 50% insensitive ratio) with weight pruning using one LSTM layer with 1500 units in word language modeling task on WikiText-2 dataset.

| Method | PPL w/o dual-module inference | PPL w/ dual-module inference |
|---|---|---|
| Dense | 85.52 | 86.21 |
| 80% weight sparsity | 86.42 | 88.46 |
| 90% weight sparsity | 88.75 | 90.96 |

---

[4]From https://github.com/NervanaSystems/distiller

