# OpenReview forum: "Dual-module Inference for Efficient Recurrent Neural Networks"
_ICLR.cc/2020/Conference — Reject_

### Official Review · AnonReviewer1 · 2019-10-16
**Official Blind Review #1**

**Rating:** 6

**Review:**

This paper attempts to compress the networks so as to accelerate the running procedure as well as save the storage. The authors propose a dual-module that is composed of a little module and big module. The big module use the full original data and parameters whereas the little module use small data and parameters by random projecting on the original ones. Through a statistical investigation, the authors provide a method to choose the little or big module dynamically. By applying this method on LSTM and GRU, the authors make them more efficient. Experimental results validate this point.

Overall, I think this paper is well written and easy to follow. The idea of dual-module is interesting and wise. The experiments is valid. However, I would like the authors to answer me two questions.
1 How do you draw Figure 2?
2 Why is the distribution of the outputs of sigmoid not bipolar? Intuitively, the distribution should be similar to that of tanh since their functions are similar.


**Experience Assessment:**

I have read many papers in this area.

**Review Assessment: Checking Correctness Of Derivations And Theory:**

I carefully checked the derivations and theory.

**Review Assessment: Checking Correctness Of Experiments:**

I assessed the sensibility of the experiments.

**Review Assessment: Thoroughness In Paper Reading:**

I read the paper thoroughly.

---

> ### Author Response · Authors · 2019-11-13
> **Response to Review #1**
>
> Thank you so much for your positive feedback on our work. Here are our answers to your questions.
> (1)	Figure 2 shows the dynamic region distribution of neuron activations, i.e., outputs of gates in LSTM. The x-axis consists of the first one hundred neurons and the y-axis is across timesteps. The left and right are of two different inputs. The key message is that the locations of neurons in the insensitive region are dynamically changing across timesteps and inputs.
> (2)	The distribution of the input gate values is from a standard LSTM with 1500 units trained by 40 epochs on PTB, following the PyTorch word language modeling example. Please note that our method does not rely on the unipolar or bipolar distribution of outputs; the criterions to search the insensitive region can cover both cases as in Equation (5).

---

### Official Review · AnonReviewer2 · 2019-10-23
**Official Blind Review #2**

**Rating:** 3

**Review:**

This manuscript proposes an approach to reduce memory access and computation in Recurrent Neural Networks.  Specifically, they train a second "little" neural network to approximate a pre-trained "big" network and use simple rules to switch between the little and the big network.  The approach can provide some speedups while reducing the total number of memory accesses and the computational cost in exchange for a mild decrease in predictive performance.

While this manuscript proposes a reasonable contribution, it lacks real comparisons to many of the common competing methods that hinder the interpretation.

Weight sparsity/pruning is a very common approach that has shown the ability for larger speedups than what is shown here.  I disagree with the assessment that "those methods require extensive retraining via regularization." Realistically, you can take a pretrained model and add the penalties with mild re-training and extensive reuse of code.  The result is also simpler to implement.  I would argue that this is less work than the proposed approach, which requires switching rules and a second trained network.  I don't know which is better, but the authors should actually evaluate whether their approach improves over the more popular approach.

The authors should give better discussion and motivation on the random projections.  This is an area with very deep theory, yet the rules are provided without a rationale.  Realistically, where does the sparsity level in the random sparse matrix come from?  Why use the rule for k in (3)?  The authors should motivate and discuss this section more.

Also note that there is existing literature on learning multiple models and switching between them, for example:
Bolukbasi, Tolga, et al. "Adaptive neural networks for efficient inference." Proceedings of the 34th International Conference on Machine Learning-Volume 70. JMLR. org, 2017.
As there is a lot of similarity in the motivations, you should discuss that line of research in your related work.

**Experience Assessment:**

I have read many papers in this area.

**Review Assessment: Checking Correctness Of Derivations And Theory:**

I carefully checked the derivations and theory.

**Review Assessment: Checking Correctness Of Experiments:**

I carefully checked the experiments.

**Review Assessment: Thoroughness In Paper Reading:**

I read the paper thoroughly.

---

> ### Author Response · Authors · 2019-11-13
> **Response to Review #2**
>
> Thank you for your valuable suggestions and comments. We have made revisions accordingly.
> (1)	Our approach is orthogonal to weight pruning. In fact, our proposed dual-module inference can work with weight-pruned models. Like the motivation in Bolukbasi et al. “Adaptive neural networks for efficient inference”, our approach focuses on dynamic execution which is not meant to replace but to complement model compression techniques. In Appendix A, we compare our approach with weight pruning. Firstly, compared with fine-grained, i.e., element-wise weight pruning, our method achieves better quality with practical speedup. Fine-grained weight pruning is effective in creating very sparse models that can be compressed to save memory footprint and bandwidth. However, specialized hardware is necessary to gain real speedup of computation given irregular sparsity. On the contrary, our method aims at reducing memory access in a structured way, i.e., reducing accesses to weight vectors rather than irregularly distributed elements, and can achieve wall-clock execution time reduction on commodity CPUs. Also, our method can improve the pruned models to further speed up inference by reducing memory access.
> a.	One major difference between our method and weight pruning methods is that our method focuses on dynamically reduce weight access while weight pruning methods prune weights permanently to reduce memory footprint.
> b.	We have added results in Appendix A to show that we can use dual-module inference on top of pruned models.
> (2)	Discussion and motivation on random projections. Random projection is a common technique for dimension reduction that preserves distances in Euclidean space. We choose the simple distribution instead of Gaussian distribution as originally proposed in Achlioptas, Dimitris (2003) “Database-friendly random projections”. The reason for using sparse random projection is that each element in the sparse random matrix can be represented in only two bits and efficiently stored in memory.
> (3)	Thank you for pointing out this related paper. We have cited and discussed in Section 5. Indeed, we share similar motivations to this paper. However, we focus on intra-layer dynamic execution while this work is on inter-layer and inter-model dynamic execution.

---

### Official Review · AnonReviewer3 · 2019-10-26
**Official Blind Review #3**

**Rating:** 3

**Review:**

In this paper, the authors design a big-little dual-module inference to dynamically skip unnecessary memory access and computation to speedup RNN inference.
It cannot only mitigate the memory-bound problem to speedup RNN inference but also leverage the error resilience of nonlinear activation functions by using the lightweight little module to compute for the insensitive region and using the big module with skipped memory access and computation. They also conduct several experiments to evaluate their approaches.


Strength:
(1)	Well written in general.
(2)	Contributions clearly stated and justified

A couple of minor questions:

(1)	The organization in Section 3 can to be improved. It is better if the author can give a brief overview of their method first and then go into details.
(2)	Some of the technical details necessary for understanding the soundness of the techniques are either missing or are poorly explained. For example, in Section 3
a.	the authors did not mention how to construct the HH module
b.	the authors did not provide detailed information of how to conduct dimension reduction since this will affect the performance
c.	many mathematical notations and equations need to be revised to increase the readability. For instance, there is no information in the paper that explain why the authors design functions in such certain way (such as equation (2) and (3))
d.	the authors did not provide enough detailed information about how to select the quantization methods since there are lots of approaches such as static (uniform) or dynamic quantization, where different methods may have different impacts on the final performance
e.	the authors mentioned that they have tried both sigmoid and tanh activation function to find the sensitive region. However, they do not provide enough reason to do so, how about using other non-linear activation functions

(3)	The organization in Section 4 can to be improved. It is better if the author can introduce the motivation of each experiment.

(4)	Parameters of the evaluation are unclear or missing. For example,
a.	what is the data size, what is the dropout, learning rate, how many time stamps for the RNN modules
b.	why the authors only use single-layer LSTM and why they select 750 and 1500 hidden units in the experiments

(5)	While the authors have applied their models on other existing method, they do not provide good discussion of results and such model seems old (released in 2016). It would be great if this approach can also be applied on other newly released models
(6)	Some tables need to be reorganized. For instance, for table 6, there needs some space between the title and the table.
(7)	While the paper has good coverage of the prior work, I do suggest the authors can also cite or discuss some newly designed models (in 2019).


**Experience Assessment:**

I have published one or two papers in this area.

**Review Assessment: Checking Correctness Of Derivations And Theory:**

I assessed the sensibility of the derivations and theory.

**Review Assessment: Checking Correctness Of Experiments:**

I assessed the sensibility of the experiments.

**Review Assessment: Thoroughness In Paper Reading:**

I read the paper at least twice and used my best judgement in assessing the paper.

---

> ### Author Response · Authors · 2019-11-13
> **Response to Review #3**
>
> Thank you for your valuable suggestions and comments. We have revised our submission based on your feedback. Our answers to your questions can be found below.
> (1)	Reorganized Section 3 with an overview subsection and a summary subsection.
> (2)	Technical details:
> a.	How to construct the HH module. The HH module, i.e., high dimension and high precision module is the original module from pre-trained models, e.g., one LSTM layer. We construct a low dimension and low precision module — the LL module — to approximate the HH module outputs but using much less overhead.
> b.    Detailed information on how to conduct dimension reduction. When constructing the LL module, we use random projection to reduce the dimensionality of inputs by multiplying the original input with a non-square random matrix as in Equation (2) and correspondingly reduce the number of parameters of the LL module. We choose sparse random projection with a sparse projection matrix that is memory efficient as the matrix elements can be represented in only two bits.
> c.  The choice of Equation (2) and (3) are from Achlioptas, Dimitris (2003), “Database-friendly random projections”.
> d. Detailed information on how to select the quantization method. We apply retraining-free uniform quantization in the LL module to keep memory overhead small. Although the uniform quantization is simple, it is good enough to reduce the precision of the little modules without degradation on inference quality (see Table 6). One explanation is that the LL module is only used in the insensitive region that can tolerate errors to a great extent. This error resilience of non-linear activation functions has been discussed in Section 2. Of course, other advanced quantization methods are allowed in our framework to further reduce the quantization error.
> e.	Although using other non-linear activation functions are possible, we restrict our exploration to sigmoid and tanh that are commonly used in LSTM/GRU-based models.
> (3)	Reorganized Section 4 by introducing the motivation of each experiment.
> (4)	Parameters in the evaluation:
> a.	Data size, dropout, learning rate, and timestamps. Our evaluation for single-layer RNNs is adapted from PyTorch’s word language modeling example, where the dataset has 10K tokens. We do not use dropout when training the LL module(s); the starting learning rate is 5, and we decay it by 4 if no loss descent has been seen on the validation dataset. The RNNs used in language modeling have 35 timestamps; the maximum generated sequence length in GNMT is 80.
> b.	Besides single-layer LSTM and GRU, we also evaluate four-layer stacked LSTMs as in GNMT. For GNMT experiments, we use the same set of parameters when training the base model as in added Footnote 2. For language modeling, we choose 1500 hidden units following the word language modeling example, and we compare our method which dynamically reduces 50% of weight accesses to the static case where only 750 hidden units are used. As we discussed in Table 2 and in Section 4.1, even at the same level of memory access reduction, our dual-module inference method can achieve better model quality.
> (5)	Although we consider fundamental sequence modeling, our method is easy to be applied to other models.
> (6)	We have revised Table 6.
> (7)	We have cited and discussed some newly designed models in Section 5.

---

### Decision · Program_Chairs · 2019-12-19

**Decision:**

Reject

**Comment:**

This paper presents an efficient RNN architecture that dynamically switches big and little modules during inference. In the experiments, authors demonstrate that the proposed method achieves favorable speed up compared to baselines, and the contribution is orthogonal to weight pruning.
All reviewers agree that the paper is well-written and that the proposed method is easy to understand and reasonable. However, its methodological contribution is limited because the core idea is essentially the same as distillation, and dynamically gating the modules is a common technique in general. Moreover, I agree with the reviewers that the method should be compared with more other state-of-the-art methods in this context. Accelerating or compressing DNNs are intensively studied topics and there are many approaches other than weight pruning, as authors also mention in the paper. As the possible contribution of the paper is more on the empirical side, it is necessary to thoroughly compare with other possible approaches to show that the proposed method is really a good solution in practice. For these reasons, I’d like to recommend rejection.